# Efficient Development of Integrated Lab-On-A-Chip Systems Featuring Operational Robustness and Manufacturability

**DOI:** 10.3390/mi10120886

**Published:** 2019-12-17

**Authors:** Jens Ducrée

**Affiliations:** FPC@DCU—Fraunhofer Project Centre for Embedded Bioanalytical Systems at Dublin City University, School of Physical Sciences, Glasnevin, Dublin 9, Ireland; jens.ducree@dcu.ie; Tel.: +353-1-700-7658

**Keywords:** Lab-on-a-Chip, microfluidic platform, functional integration, technology readiness level, standardisation, Design-for-Manufacture, scale-up of manufacture, tolerance-forgiving design

## Abstract

The majority of commercially oriented microfluidic technologies provide novel point-of-use solutions for laboratory automation with important areas in the context of the life sciences such as health care, biopharma, veterinary medicine and agrifood as well as for monitoring of the environment, infrastructures and industrial processes. Such systems are often composed of a modular setup exhibiting an instrument accommodating rather conventional actuation, detection and control units which interfaces with a fluidically integrated “Lab-on-a-Chip” device handling (bio-)sample(s) and reagents. As the complex network of tiny channels, chambers and surface-functionalised zones can typically not be properly cleaned and regenerated, these microfluidic chips are mostly devised as single-use disposables. The availability of cost-efficient materials and associated structuring, functionalisation and assembly schemes thus represents a key ingredient along the commercialisation pipeline and will be a first focus of this work. Furthermore, and owing to their innate variability, investigations on biosamples mostly require the acquisition of statistically relevant datasets. Consequently, intermediate numbers of consistently performing chips are already needed during application development; to mitigate the potential pitfalls of technology migration and to facilitate regulatory compliance of the end products, manufacture of such pilot series should widely follow larger-scale production schemes. To expedite and de-risk the development of commercially relevant microfluidic systems towards high Technology Readiness Levels (TRLs), we illustrate a streamlined, manufacturing-centric platform approach employing the paradigms of tolerance-forgiving Design-for-Manufacture (DfM) and Readiness for Scale-up (RfS) from prototyping to intermediate pilot series and eventual mass fabrication. Learning from mature industries, we further propose pursuing a platform approach incorporating aspects of standardisation in terms of specification, design rules and testing methods for materials, components, interfaces, and operational procedures; this coherent strategy will foster the emergence of dedicated commercial supply chains and also improve the economic viability of Lab-on-a-Chip systems often targeting smaller niche markets by synergistically bundling technology development.

## 1. Introduction

Since their inception between the late 1970s and early 1990s [1,2,3], the field of microfluidics has tremendously advanced from creating sheer miniaturised versions of conventional pumps, valves and analytical equipment to harnessing (and managing) specific, typically micro-confinement-related effects that enable a broad repertoire of novel applications. As fluids, in particular bioliquids, are at the very core of all living matter, applications are predominantly found in the broader context of the dynamically emerging life sciences—for instance, in biomedical point-of-care diagnostics, at-line bioprocess monitoring or as portable, widely autonomous solutions for decentralised monitoring of infrastructures, industrial processes and the environment.

There have been numerous surveys analysing the commercial potential of microfluidics-enabled products [4,5], indicating a present market in the region of US$ 10 billion, with persistently strong annual growth rates [6]. The lion’s share of this market is related to microfluidics-enabled products for decentralized, fully automated preparations and/or (bio-)analytical testing as addressed in this work [7].

Conventional bioanalytical methods typically require rather complex and expensive liquid handling robotics and/or well-trained professional staff. This work considers a significant subset of micro-fluidic technologies for automating and parallelising liquid handling protocols comprising sample preparation, reagent management and/or detection as the backbone of common bioanalytical laboratory procedures.

Such technologies frequently involve a modular setup made of a single-use polymer microfluidic chip interfacing with a reusable instrument. While the development of instrumentation can, for the most part, resort to conventional units for actuation, detection, interfacing and software, the engineering of the chip itself poses particular challenges as its network of submillimetre-scale fluidic channels and chambers including bonded lids, surface coatings, micron-scale features and stored (dry or liquid) reagents can, in the vast majority of possible use cases, not be (fully) recovered. Successful commercialisation of microfluidic technologies is thus tightly coupled to the operational performance, robustness, development and manufacturing cost of the disposable, usually a microstructured and fluidically sealed polymeric chip.

## 2. Important Market Driver: Fluidic Integration

The initial hype around micro-electro-chemical systems (MEMS) [8] and later microfluidics was rooted in the unparalleled success stories of microelectronics; even after five decades, the drastic increase in integration density continues to follow the breathtaking trajectory of Moore’s law [9] that has empowered a series of game-changing technological breakthroughs like personal computers, smartphones, the internet, big data and artificial intelligence. A key driver of microelectronics remains progress in integration by miniaturisation towards nano-scale structuring to “cram more components” [9] on a given piece of real estate.

The value of Lab-on-a-Chip systems to the end user will also increase with the density of functional integration, parallelisation and automation, while assuring satisfactory performance, user convenience, reliability and costs. Yet, there is a set of unique issues to be addressed regarding such efforts.

## 3. Challenges for Technology and Business

### 3.1. Miniaturisation

While MEMS devices, in particular in the area of physical sensing and communication, have truly pervaded mass consumer markets by now, fundamental laws of physics dictate that systems based on mechanical principles cannot be scaled down to the same extent (without a fundamental change in their operational principles) as electronics where there still seems to remain some significant “room at the bottom” [10]. Presently, cost reduction and packaging seem to be amongst the main industry drivers for MEMS.

While it is a prerequisite for higher-level functional integration of Lab-on-a-Chip applications, miniaturisation is generally considered to hit technological boundaries even earlier than MEMS; one of the fundamental reasons for this restraint is constituted by basic hydrodynamics leading to a massive reduction of the flow rate with (the inverse square of) the channel cross section and the related technical challenge of its fine control; other severe roadblocks might be hitting the limits of detection for shrinking (sample) volumes (including the risk of not having or finding any target molecule in solution, especially when combined with low concentration), and possibly also the adversely increasing impact of surface-related effects.

### 3.2. Life Science Engineering

For microfluidic devices addressing applications in the life sciences, the complexity and intrinsic variability of bioliquids, i.e., sample and reagents, and the intricate interplay of fluid dynamics, materials, manufacturing processes and detection schemes (Figure 1) represent a significant challenge for engineers; other than many conventional systems, hydrodynamics in microconfinement is governed by large surface-to-volume ratios, and so the bio-physico-chemical properties of inner walls and other solid-liquid-gas interfaces, which are often hard to define and stabilize over time, assume a dominant role in flow control and reaction kinetics. While these effects might be technology exploited, their intrinsic, poorly defined variations need to be carefully considered and managed for arriving at viable end-user products featuring superior performance paired with sufficient reliability at competitive pricing.

### 3.3. Higher-Level System Integration

Full point-of-use automation and parallelisation of life-science protocols on a real-world Lab-on-a-Chip device will require high operational robustness on a system level. To illustrate the tremendous challenge, we assume that a (laboratory) procedure can be split up into N (for the sake of simplicity independent) functional elements, such as laboratory unit operations (LUOs) or interconnecting valves, with each of them displaying a reliability of X; the overall reliability of the system will thus simply be calculated as $\Gamma =X^{N}$; so for N=10 assay steps, each characterised by X=98%, we obtain Γ≈82%, which would be rather poor in most end-user contexts; the same system-level reliability would already require X≈99% for increasing the integration density to N=20.

This example has obviously been somewhat oversimplified to illustrate the trend and dire need to assure maximum reliability for each functional module, in particular considering the complex multi-disciplinary correlations portrayed in Figure 1 and the innate variability of surfaces and bioliquids. Alternative approaches to tackle critical, system-level reliability are rugged, one-pot or single-step assays, if commensurate with the often rather stringent demands on quality and timing of sample preparation and detection.

### 3.4. Business Landscape

In addition to these significant technical challenges, the business-to-business landscape for Lab-on-a-Chip technologies still dwells in its infancy compared to more widely commercialised conventional technologies [11]. In recent decades, an expanding, but still rather small number of companies specialised in contract manufacturing of microfluidic systems.

Somewhat paradoxically, a lot of these foundry-type organisations are sourced to accompany the research and technology development (RTD) phase, while especially large companies often eventually decide to internalise larger-scale production. Microfluidic foundries would also regularly stress that they are only getting involved in later stages of the product development when it is often difficult to still accommodate design changes imposed by their in-house portfolio of fabrication schemes.

Notably, also a critical mass of providers for designated manufacturing equipment, materials, components, software and professional services as sentinels for mature supply chains is still missing, possibly also owing to the huge diversity of underlying technologies and their application scenarios. Whether cause or reason, there is a decisive fragmentation in the field of microfluidics, so many commercial initiatives choose an in-house approach to develop microfluidic design and set up manufacture for their life-science applications, thus discarding the well-known synergies of specialisation in task-sharing economies.

In this regard, compliance with existing or the promotion of specialised standardisation would certainly facilitate the emergence of effective supply chains for materials, components and services to coherently coordinate characterisation and validation procedures as well as for chip-internal and peripheral interfaces.

## 4. Strategy towards High Technology Readiness Levels (TRLs)

We will outline here a strategy towards reaching high technology readiness levels (TRLs) that may eventually contribute to a new type of design and foundry service addressing the peculiar challenges of microfluidics-enabled automation for multiplexed sample preparation and testing of biosamples at the point of use. The manufacturing-centric methodology outlined here aims to de-risk and expedite cost-efficient advancement of solutions towards high TRLs. The success of this approach is tightly linked to the central paradigms
Design-for-Manufacture (DfM) andReadiness for Scale-up (RfS),
and to accounting for tolerances in properties of
Materials and surfaces,Geometrical dimensions and features, andBioliquids
by scrutinising, factoring in or managing, as far as possible and primarily through engineering solutions, their impact on (quantitative) Key Performance Indicators (KPIs) of the Lab-on-a-Chip system regarding flow control/profile/rates, LUOs, preparations and bioassays.

This systematic approach may be accompanied by simulation and experimental validation as well as well-proven methods such as Failure Mode and Effects Analysis (FMEA).

### 4.1. Operational Robustness

In addition to a market-competitive price, successful commercialisation indispensably requires highly repeatable operation and consistent performance. Such reliability constitutes a particular challenge for microfluidic systems which are typically prone to sizeable tolerances and artefacts in front- and backend manufacturing processes [12], like injection moulding, hot embossing, bonding, and surface modification, and also tend to very sensitively respond to the physico-chemical and biological cues at the complex crossroads between engineering and the life sciences (Figure 1).

The performance optimisation evaluates tolerances of the input parameters for the Lab-on-a-Chip system like the physico-chemical properties and dimensions of manufacturing, materials, instrumentation, (bio-)liquids and ambient conditions, and analysis how they, immediately or indirectly, translate into the KPIs of flow control, sample preparation and transduction/detection schemes; their combined effect then determines the KPIs of the target application (Figure 2). Methodically, the mostly empirically guided optimisation cycles of the bioassay should only be carried out after the robustness of fluidic operation is assured with pilot series of the Lab-on-a-Chip devices.

The microfluidic systems considered in this article are assembled from a repertoire of components such as (structured) plastic parts, films for bonding/sealing and inserts like stationary phases or membranes acting as filters or diffusion barriers, respectively. According to the considerations in the previous section, system-level reliability rapidly drops with increasing counts and tolerances of its constituents. Already from a mere cost and reliability perspective, the number of parts to be assembled should be minimised; further strategies to enhance reliability are the avoidance of largely varying forces, with hard-to-define properties of surfaces and bioliquids as essential parts of the operational principles, and, simultaneously, to engineer tolerance-forgiving designs. Vice versa, any tolerance-forgiveness of the bioassay protocol and detection scheme would further increase reliability.

Bubble formation and trapping represent other common failure modes [13,14], in particular during priming of multi-branched microfluidic networks. Solutions often involve degassing of liquids or capillary guides to impose reproducible, bubble-free priming of the microstructured network with liquids. Due to their drastically diverging mass density, gravitational effects have also been harnessed to remove unwanted bubbles from the liquid phase.

### 4.2. Design-for-Manufacture (DfM)

While high manufacturing precision boosts reliability, it also tends to notably drive up costs of product development and production; therefore, a reasonable sweet spot needs to be found for manufacturing tolerances by proper fluidic design.

In general, any “extreme” requirements on the accuracy and precision of dimensions (e.g., channel widths and depths better than 10 µm) and fidelity of shapes, e.g., on the thickness variation across the (macroscopic) chip as well as on the definition of geometrical features such as the sharpness of corners and edges, may substantially increase fabrication costs; capillary pressures resulting from the interplay of structure and contact angle are also hard to accurately set—on the one hand, due to innate variability of the interfacial energy of the biosample(s); on the other hand, owing to the rather poor reproducibility and temporal stability of the surface energy over the life cycle of the device.

Considering that any parameters affecting the system-level performance of the Lab-on-a-Chip solution, e.g., arising from (unavoidable, often significant) variations in properties of materials and surfaces, also along scale-up from prototyping to mass manufacture, as well as of hydrodynamic or biochemical characteristics of liquid samples and reagents will be prone to deviations, robust, tolerance-forgiving design is paramount for achieving high reliability.

Design-for-Manufacture (DfM) considers that the maximum tolerances of each fabrication scheme represented by vertical pillars in Figure 3 will not compromise meeting the targeted KPIs set out for fluidic operation and the bioassay.

### 4.3. Readiness for Scale-Up (RfS)

A set of different manufacturing schemes will have to be sourced for application development of many bioanalytical Lab-on-a-Chip systems for the following reasoning:The number of Lab-on-a-Chip devices required for statistically sound testing and optimisation of bioanalytical performance will range from the 100s to 1000s, thus notably exceeding numbers that can reasonably be supplied through prototyping by at least 1–2 orders of magnitude.Due to the outlined, very complex interdependencies of biosamples with the Lab-on-a-Chip, such testing needs to be carried out on devices exhibiting bio-physico-chemical properties that are as close as possible to the technology for eventual larger-scale manufacture.Related high-throughput manufacturing schemes overwhelmingly involve tool-based polymer replication, e.g., injection moulding (or hot embossing). Their high upfront cost of tooling and process optimisation often rules out frequent design iterations.So, despite causing a substantial risk of discontinuity in functionally critical device properties along with the scale-up of device manufacture, the time and cost of tool-based replication schemes often force developers to employ prototyping techniques already during the initial design iterations. This structuring step is usually based on direct-writing schemes such as precision milling, laser ablation, knife cutting and/or 3D printing, or replication schemes such as hot embossing, which exposes the material to different processing conditions.

Consequently, the tolerances required for meeting the operational KPIs for fluidic and assay performance have to be met over the entire range of manufacturing scale-up from initial prototyping by direct patterning techniques like precision milling or 3D printing, to tool-based smaller- and larger-scale replication, assembly and possible backend processes. Additionally, DfM has to be taken into account consistently along all stages of scale-up (Figure 3); for instance, if proper demoulding of downstream replication by injection moulding requires draft angles on the sidewalls, corresponding geometries should already be included in the prototyping stage, e.g., to assure comparable fluidic performance. We refer to this cross-sectional manufacturing compliance as Readiness for Scale-up (RfS).

Adhering to these DfM and RfS paradigms as well as systematic implementation of common methodologies like Failure Mode and Effects Analysis (FMEA) helps decrease overall costs and time scales of product development towards elevated TRLs, and thus significantly de-risks investment in RTD. Within the platform strategy outlined in the following section, this sizeable upfront effort can be efficiently recycled for future projects, in particular when underpinned by proper documentation through Standard Operating Procedures (SOPs) and Design History Files (DHFs), which would also assist regular approval.

## 5. Platform Approach

### 5.1. Motivation and Lessons Learnt

All living nature relies on the same set of basic biomolecular building blocks such as nucleic acids and amino acids/proteins, which are contained in or interact with cells; superordinate, multicellular organisation leads to the formation of tissues, organs and eventually complex organisms. On a macroscopic scale, groups of species such as quadrupeds display the same basic plot in terms of the arrangement of their (four) legs, tail and head featuring two eyes, a nose, ears and mouth along their central torso.

Similarly, the success of many mature industries derives from the capability to rapidly configure and customise a broad repertoire of products and associated user experiences from a core set of modules; such technology platforms exhibit a widely intra-compatible set of materials, components, interfaces, processes, services and test methods. Prominent examples for such platforms are automotive and microelectronics industries as well as personal computers, smartphones and their operating systems, data interfaces and communication protocols, peripherals, development tools and their means of production.

In the life sciences, standard microscope slides or liquid handling automation represent such a platform where the 75 mm × 25 mm (or ”3 × 1”) footprint and the common 96-, 384- and 1536- well plate formats, respectively, are shared between suppliers of instrumentation for liquid handling automation and readout as well as consumables and software.

### 5.2. Integrated Microfluidic Platforms

Here, we seek to adopt essential lessons from successful industries while still accounting for the specific challenges within life science engineering. In this sense, we define a microfluidic platform by a repertoire of design, manufacture and instrumentation technologies from which, through proper choice of complementary bioassay and detection schemes, manifold applications may readily be derived (Figure 4). The objective of this holistic platform approach is to swiftly enhance TRLs, i.e., the reliability and manufacturability of microfluidics-enabled solutions while keeping costs, risks and time scales for development of product and production technologies at bay.

In this work, we focus on microfluidic technologies for automating typical in vitro procedures in life-science laboratories. From an engineering point-of-view, such protocols—commonly comprising multi-step and multi-reagent sample preconditioning for downstream detection—can be interpreted as a sequence of laboratory unit operations (LUOs) which manipulate the physical and/or (bio-)chemical properties of the sample.

Conventionally, manual pipetting or liquid handling robotics transfer and condition the sample and reagents between different containers and instruments where LUOs are carried out. The integrated microfluidic systems discussed here integrate these processes on a single-use chip which features flow control elements for pumping, valving and routing between the microfluidic equivalents of conventional LUOs; this Lab-on-a-Chip is operated by an instrument providing actuation and readout. As reasoned above, preferential schemes for LUOs and flow control are amenable to the DfM and RfS paradigms and tend to be widely independent of often spatio-temporally poorly defined properties of bioliquids and surfaces such as viscosity or surface tension, and resulting capillary pressures induced by corners and channel walls.

A particular challenge for common, real-world Lab-on-a-Chip systems geared for user-friendly point-of-use testing is longer-term storage and on-demand release of liquid and dry (bio-)reagents; such chip-based reagent storage faces manifold bio-physico-chemical issues along its life cycle extending from loading at the factory to transport, shelf storage and handling by the end user; amongst them are evaporation, material absorption, (bio-)chemical stability and reliable release of well-defined liquid volumes upon activation at the point of use. Amongst the proposed solutions are vapour-impermeable blister pouches [15,16], stick packs, glass ampoules, films or coatings enclosing the reagent. Opening might be realised by mechanical, e.g., manual or externally powered removal or perforation of the barrier material.

Prominent examples of integrated microfluidic platforms where flow control elements coordinate the spatio-temporal arrangement of a range of LUOs towards sample-to-answer automation are based on rotationally induced centrifugal fields [17,18,19,20,21,22,23,24,25,26], electrokinetics [27], acoustophoresis [28,29,30,31], and digital (droplet) microfluidics on electrowetting-on-dielectric (EWOD) [32,33,34,35,36], surface acoustic waves (SAWs) [37,38,39] and multiphase flows [40,41,42], possibly supported by externally actuated precision pumps and mechanical valves.

## 6. Commercial Dimension

### 6.1. Rapid Configurability

There is a range of bioassay formats implemented on common microfluidic systems: general chemistry, immunoassays, nucleic acid testing and identification/counting of bioparticles such as cells, bacteria and extracellular vesicles.

To leverage the underlying concept, the platform hosts a comprehensive library of configurable modules for flow control and LUOs which are geometrically parametrised and abide by the DfM and RfS guidelines, thus allowing rapid customisation for new applications. This way, the technologically rather splintered arena for Lab-on-a-Chip technologies can bundle RTD efforts of smaller niche markets to still efficiently tap into economically important economy-of-scale effects.

### 6.2. Standardisation

Standards represent a signature of a task-sharing economy, whether they are open, proprietary and/or confidential, and thus exclusive for use within certain entities, corporations or consortia, or subject to licence fees. They may be issued by a regulatory body, but also internally established by a consortium or organisation. In the context of the platform concept discussed here, such standards may define properties of materials, processes, surfaces, geometries, tolerances, interfaces and validation procedures.

Several standardisation initiatives have been suggested and launched within the microfluidics community [43,44,45,46], e.g., to assist and expedite regulatory approval. However, standards have also been blamed for hampering technological innovation, especially if introduced too early in the development.

While RTD organisations may choose not to be fully compliant, e.g., for organisational reasons, their product development may still be informed and guided by industry standards for quality management such as ISO: 9001, ISO: 13485 and the 6σ framework.

### 6.3. Supply Chains

Sentinels of mature, task-sharing economies are supply chains where independent, specialised stakeholders systematically collaborate on bringing a product or service from supplier to customer. Such supply chains comprise organisations, people, activities, information, and/or resources. Formal or de facto standards assure compatibility, interoperability, safety, repeatability or quality to coordinate seamless transfer of materials, components, systems, data and services based on jointly agreed characterisation and validation methods. Lab-on-a-Chip initiatives could be largely boosted by the establishment of such designated supply chains involving providers for microfluidic design, chip manufacture, bioassay development and instrumentation, where the platform, in the concept described above, defines the standards.

## 7. Summary and Outlook

While microfluidic technologies have by now been developed over approximately three decades, true killer apps have not surfaced. Yet, there is still tremendous research and commercial potential in numerous application areas, often associated with spatial and functional integration and automation of life-science procedures, e.g., for sample preparation and detection of biosamples in decentralised “point-of-use” scenarios.

A key technological challenge represents the innate diversity of the biosamples and the high sensitivity of microfluidic effects to surface properties in combination with the typically disposable nature of the Lab-on-a-Chip devices; while operational reliability is associated with the high accuracy and precision of input parameters and related KPIs, manufacture needs to be cost-efficient, thus, for most business-relevant applications, ruling out pricy materials and processing techniques. Critical, system-level operational robustness of larger-scale integrated Lab-on-a-Chip systems needs to be underpinned by tolerance-forgiving fluidic designs and robust assay schemes.

The usually sizable spread of bio-physico-chemical properties of samples and reagents also makes comprehensive data collection imperative for arriving at sufficient statistics; in turn, this requires the availability of larger amounts of chips, even during the development phase.

To minimize risks associated with changes in technology and to support subsequent regulatory approval, these pilot series should be manufactured with materials, geometries and processes as close as possible to the eventual mass fabrication schemes such as injection moulding, as the latter involve significant setup times and costs for tooling and optimisation of the replication process.

Another roadblock towards commercialisation is that individual microfluidic solutions often address niche markets, thus struggling to tap into economy-of-scale effects for justifying massive upfront investment in RTD on product and production technologies. The platform approach should, therefore, be flanked by rapid configurability and standardisation to allow bundling development efforts and to encourage the formation of commercial supply chains.

This work outlines how a manufacturing-centric platform approach following the Design-for-Manufacture (DfM) and Readiness for Scale-up (RfS) paradigms efficiently bundles RTD efforts to substantially de-risk and accelerate the development of economically viable and operationally robust Lab-on-a-Chip solutions.

There is still tremendous prospect in the field of microfluidics, especially when taking into account evident synergies with rapidly emerging, disruptive 21^st^-century technologies like artificial intelligence (AI), Internet of Things (IoT), Big Data and additive manufacture/3D printing. It also needs to be considered that much of the foundational intellectual property in the space of microfluidics was filed in the 1990s, and thus has or is about to enter the public domain to clear the road for new players to stimulate the field.

## Figures and Tables

**Figure 1 micromachines-10-00886-f001:**
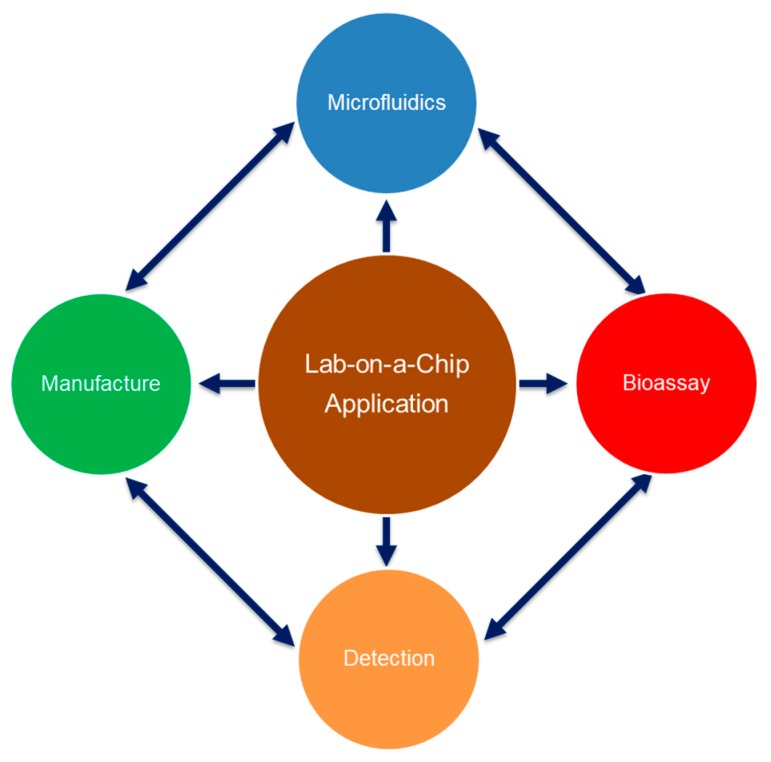
The functionality of Lab-on-a-Chip application for the life sciences of microfluidics, manufacture, bioassays and detection technologies, which may mutually impact each other, and thus the overall Key Performance Parameters (KPIs) of the device in a highly sensitive manner. These four elements thus form a technology complex that needs to be comprehensively mastered to arrive at commercially viable solutions.

**Figure 2 micromachines-10-00886-f002:**
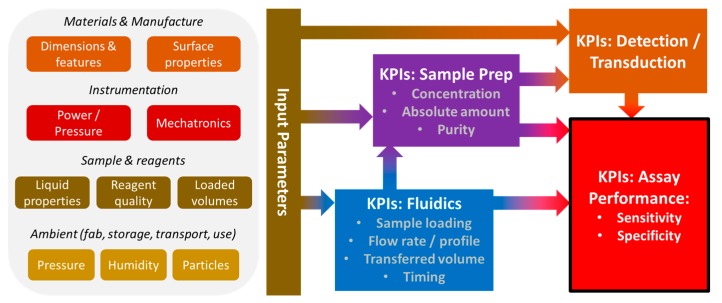
Typical (simplified) causality chain between tolerances of the input parameters for the Lab-on-a-Chip module and the performance of the target application, e.g., the quantitation of a biomarker in the loaded sample. These input parameters are set by the device materials, manufacturing, instrumentation, sample, reagents and ambient conditions, which immediately or indirectly impact Key Performance Indicators (KPIs) of flow control, sample preparation and transduction/detection to eventually determine the sensitivity and specificity of the assay.

**Figure 3 micromachines-10-00886-f003:**
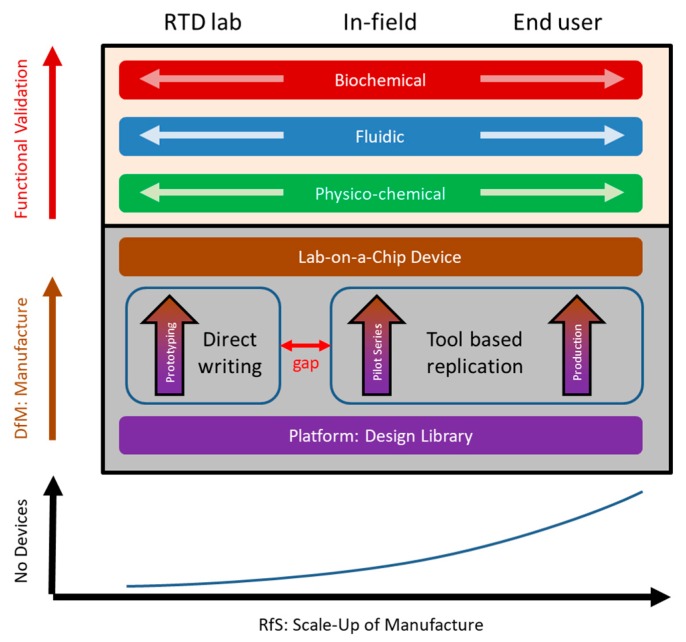
The manufacturing-centric approach involves the three pillars: prototyping, pilot series and mass manufacture related to the lab-based research and technology development (RTD) phase, in-field testing and application by the end user, respectively. Each of these vertical strands converts an initial design into a fully assembled device that is subsequently validated according to the specified quantitative Key Performance Indicators (KPIs) for its physical/geometrical properties, fluidic operation and biochemical functionality along the laboratory protocol to be automated. Design-for-Manufacture (DfM) assures that the concept can be manufactured while meeting quantitative KPIs along each vertical path, separately; Readiness for Scale-Up (RfS) concerns the consistency of KPIs when advancing “horizontally” through the scale-up of manufacture. Importantly, there is a critical gap between direct writing-based prototyping and manufacturing larger numbers involving tool-based replication, e.g., by injection moulding or hot embossing; so the prototypes should mimic as much as possible the structural and bio-physico-chemical features of the devices obtained in subsequent stages of scale-up.

**Figure 4 micromachines-10-00886-f004:**
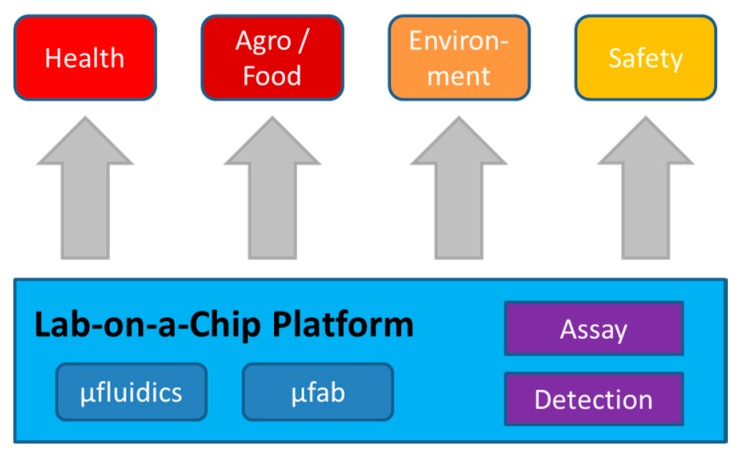
Based on the microfluidic platform comprising a geometrically parametrised library of LUOs and flow control elements and common materials and fabrication schemes, technological solutions for point-of-use automation of common laboratory procedures can rapidly and efficiently be generated for manifold applications in the life sciences.

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
