# Peer review of "Efficient Development of Integrated Lab-On-A-Chip Systems Featuring Operational Robustness and Manufacturability"

_micromachines, 2019, doi:10.3390/mi10120886_

Round 1
Reviewer 1 Report
The manuscript authored by Jens Ducrée presents an excellent and critical overview of all critical aspects that scientists working in the microfluidic/biosensing arenas must consider to potentially bring these combined fields closer to a long promised societal impact. Overall, the manuscript reads very well and all key concepts are very clearly organised, certainly being of high interest to a very broad community of readers. I have just a few minor comments/suggestions, listed below, which in my view can improve the work even further.
1- The paragraphs in lines 20-25 and 29-34 (Abstract) summarise excellent points covered in the manuscript but the phrases are too long, making the text harder to read and for the key points to come across to the reader.
2- Typo in line 63 (commercialisation).
3- Section 2 (Important market driver: Fluidic integration) could be an introductory part of section 3 (Challenges for technology and business) or a first subsection before 3.1.Miniaturization, rather than a short standalone section.
4- Typo in line 76 (there is "a" set of).
5- Line 88. I would say the main issue is more related to the massive increase in fluidic resistance and lack of fine control of the flow rate (without a massive increase in cost), making the handling and integration much more challenging, rather than the flow rates being low. It is possible to effectively engineer fit for purpose bioassays at low flow rates if time is not a critical constraint.
6- Line 89. While obvious for many of the readers, I think it would be important to specifically mention the possibility of not finding any molecule in solution at all for low concentrations and very low volumes.
7- Line 117-120. Another research direction of optimizing traditional protocols to include less assay steps in miniaturized formats could also be highlighted (e.g. efforts to build rugged one-pot assays or single-step), since lowering the fail rate below 1% is extremely difficult or even impossible for some LUOs.
8- Lines 149-154. Flow velocity and profile could also be considered central paradigms which must have room to tolerate variability.
9- Line 170. I agree with this but bioassays can also be designed in such way to be more tolerant to less robust fluidics, which is relevant to decrease costs on the fluidics/microfab side for e.g. in resource-limited settings. So I would say it is critically important only for the fluidic parameters to be appropriately characterized in terms of precision, since any existing variability needs to be taken into account when developing the assay.
10- Line 191. Would it be possible to provide some approximate quantitative data regarding this point? What can be considered extreme for tool based replication and how fast can the costs be driven up in this case?
Author Response
The author wishes to thank the reviewer for his/her very qualified assessment of the manuscript. The author considered all comments and updated the manuscript accordingly.
The author also added an additional schematic (new Fig. 2, subsequent figure numbers and text references have been updated) to better illustrate the important correlation between input parameters and key performance parameters (KPIs) that are frequently referred to in the context of the manuscript.
Reviewer 2 Report
This concept paper describes the challenges, strategies, platform approaches, and commercialization aspects in lab-on-a-chip product development. The manuscript is well written and can provide insights to product developers in relating fields.
Analyzing a few success and failure examples in lab-on-a-chip product development will be a bonus to this manuscript.
Author Response
The author wishes to thank the reviewer for the report. Accordingly, the manuscript was revised for minor typos and grammar issues. The author also agrees that examples would certainly provide further support. Yet, also related to its novelty, the author is not aware that similar approaches have been consistently pursued and implemented; as, if at all, only partial features have been published by other research initiatives, making it very hard to concisely cover them without confusing the reader and going beyond a reasonable length for the proposed manuscript.